# Attach That There: Investigating 3D Virtual Assembly Assistants That Point Into the Real World

Category: Research

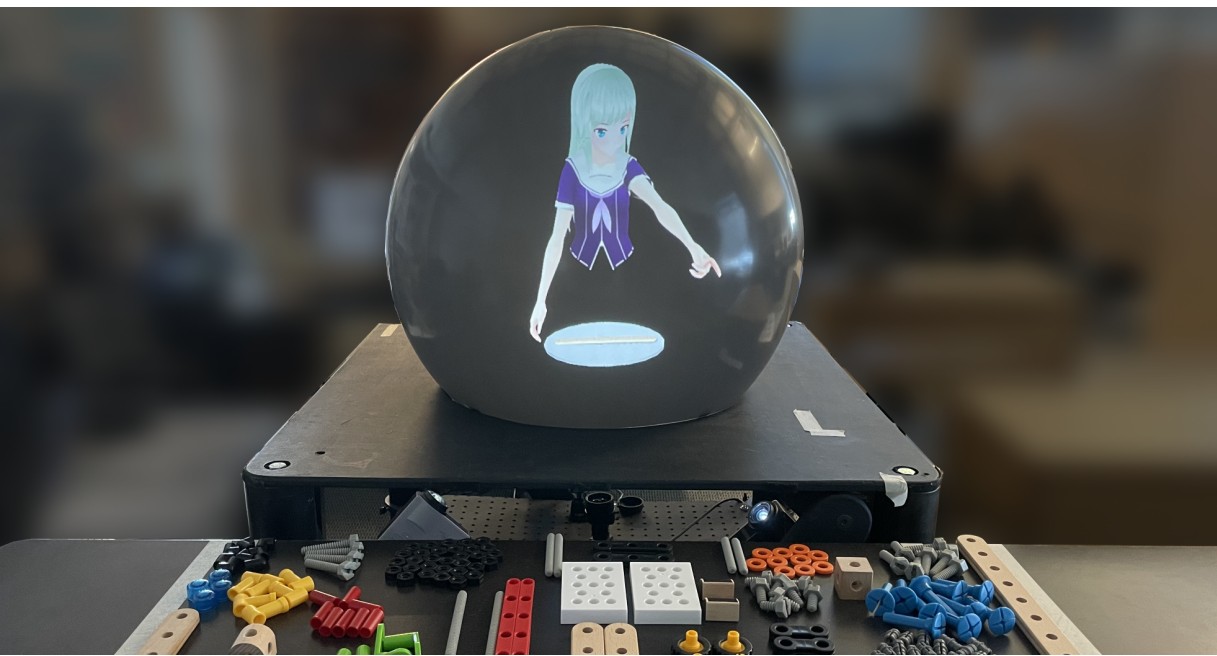

Figure 1: Assembly assistant pointing to real world targets from within a spherical FTVR display.

## ABSTRACT

Gestures are a fundamental part of human communication. However, commonly used voice assistants do not exploit the advantages of human-like nonverbal communication. We present an Embodied Conversational Agent (ECA) with the ability to explain assembly steps and point to indicate real-world targets. To enable accurate pointing into the real world, we implemented our ECA in a spherical Fish Tank Virtual Reality (FTVR) display. We evaluated the effect of a pointing ECA on the performance and experience in an assembly scenario, as well as investigated whether spherical FTVR displays provide an advantage over 2-dimensional (2D) flat displays. Results show that, while the spherical FTVR was preferred in all conditions, pointing to real pieces did not reduce assembly time or errors compared to showing virtual pieces by holding them up. Based on our findings, we provide design insights and research directions for ECAs with pointing gestures in an assembly scenario.

**Index Terms:** Human-centered computing—Human computer interaction (HCI)—Interaction paradigms—Virtual reality Human-centered computing—Human computer interaction (HCI)—Interaction paradigms—Pointing

## 1 INTRODUCTION

Spherical Fish Tank Virtual Reality Displays (FTVR) offer unique opportunities for interactions. While conventional Virtual Reality (VR) displays only support interactions in the virtual world, FTVR displays are non-immersive. Thus, they allow for pointing from within the display to the real space surrounding it, which makes them particularly suitable for implementing 3D Embodied Conversational Agents (ECAs).

Through their embodiment, ECAs have the ability to provide ad-ditional human-like nonverbal cues, like for example gestures [10]. Deictic gestures, which accompany speech, are a common method to indicate objects and guide the attention to them by substituting linguistic expressions with a pointing gesture [21]. This is particularly helpful in collaboration scenarios where establishing a mutual understanding is essential for successful communication [14]. Deictic gestures are for example used when indicating the position of an object in the room with the answer "it is over there" accompanied by a pointing gesture, instead of describing the location of the object in detail.

Deictic pointing can not only enhance the interaction in reality, it can also improve the interaction with ECAs, as they allow users or conversational agents to indicate objects they are talking about. Previous work has shown that a feature description accompanied by a deictic gesture, increases accuracy in identifying a target [4]. Moreover pointing gestures simplify the language dialog by allowing for simpler and shorter descriptions and therefore enable references in situations where descriptions alone would not be possible (e.g. when multiple similar objects are present) [21].

We believe that an ECA with the ability to point into the real world would leverage the multi-modality of human communication [32], and therefore enables more natural human agent interactions. While there are many studies on how humans perceive and use gestures, this knowledge can not directly be applied to ECAs, as there is a difference between how humans use and interpret pointing gestures [5]. Previous studies found that it is possible to implement an ECA to point into the real world with a similar or higher accuracy than a real person [35]. However, the effect of an ECA using pointing into the real world accompanied by verbal cues on the interaction experience has not been studied yet.

In this study we investigate how ECAs with pointing gestures

influence the interaction experience and performance in an assembly scenario. For this purpose, we implemented an ECA with the ability to guide users through the assembly steps by using voice instructions and gestures. To enable pointing from within virtuality into the real world, we use a spherical FTVR display for our ECA. Our spherical FTVR display adapts content to the user's viewpoint by rendering perspective corrected vision and providing motion parallax as well as stereoscopic cues to improve depth and size perception [20].

**Contributions**: 1) We created a novel virtual assistant with the ability to point into the real world, which can be modified and used in other pointing related AR/VR/XR scenarios. 2) We evaluated assembly time, errors and user preference of different display forms and ECA gestures (see Section 4). Our results show that a spherical FTVR display is preferred over a flat 2D display for an ECA. 3) Based on our study, we provide design insights and future research directions for designing ECAs with pointing gestures in assembly scenarios.

## 2 RELATED WORK

While deictic gestures are one of the most commonly used forms of non-verbal communication, there are some challenges when implementing them for ECAs. In the following, we first provide an overview over deictic pointing in human communication. Afterwards, we discuss work on how deictic gestures can be implemented in ECAs and the advantages spherical FTVR displays provide.

### 2.1 Deictic Pointing Gestures

In their everyday life, humans use deictic pointing gestures when they indicate proximal objects by extending their arm and index finger towards a pointing target. Deictic gestures are fundamental when communicating to establish a mutual understanding and help to direct attention to people or objects, especially when the use of speech only is ambiguous [30, 32]. Thus, deictic gestures are particularly suitable in an assembly scenario, where spatial deixis is important, since they can substitute certain spatial linguistic expressions and indicate objects [15].

How human gestures are interpreted is a key issue in gesture research [19]. Pointing gestures can be distinguished in proximal and distal [38]. Proximal pointing occurs when the pointer touches the target, while distal pointing occurs when the target is situated too far away and the goal is to locate the target's position in a shared environment [8]. We will focus on distant pointing, as the goal is to implement an ECA that assists in an assembly process by pointing at distant pieces. The major challenge of distant pointing is detection accuracy, which quantifies how successful observers can identify pointing targets. Bangerter et al. [5] showed that bias in pointing target detection was small for both vertical and horizontal pointing, while detection accuracy was lower for peripheral targets than for central ones.

### 2.2 Embodied Conversational Agents (ECAs) with Pointing

ECAs are virtual agents that inhibit conversational behaviors and are human-like in the way they use their bodies in conversations [12]. Cassell [12] defined ECA as having the ability to recognize, generate and respond to verbal and non-verbal input, deal with conversational functions as well as give signals that indicate the state of the conversation and contribute new propositions.

Previous research showed that the presence of an ECA can improve the interaction between the user and the agent and has a positive effect on the retainability of information, independent of the realism of the embodiment [6]. Yee et al. [26] found that agents with a visual representation lead to more positive social interactions compared to agents without a visual representation. Furthermore, they confirm previous findings that the degree of realism may matter

very little: animated highly realistic faces might appear unnatural or disturbing, which confirms Mori et al.'s [25] uncanny valley effect.

In the same way as humans use gestures, gestures can also be implemented in ECAs to enable new interaction possibilities. Previous research showed that the integration of gestures in ECAs influences the ECA's personality [9], helps to achieve a sense of co-presence [3], and improves user perceptions of friendliness and trust [32]. Research in human-robot interaction found that robots using gestures increased the user performance while decreasing perceived workload for challenging tasks [24].

### 2.3 ECAs in Different VR/XR Platforms

While considerable research has been done targeting how gestures in the virtual world are perceived and influence the interaction experience with ECAs, there is only little research on ECAs pointing into the real world. Wu et al. [34] investigated different pointing cues for ECAs pointing into the real world using a spherical FTVR display. Results show that a combination of head and hand cues yielded the best accuracy with 82.6% for fine pointing (15°) compared to hand-only or head-only cues. In a second study, Wu et al. demonstrated that an ECA using arm vector pointing can point to a physical location with comparable or even better accuracy than a real person [35]. Unlike humans who use an eye-fingertip alignment for pointing, which yields a perceptual bias [4], ECAs can be implemented using arm vector pointing to improve detection accuracy [35]. Since previous work already showed high pointing accuracy for ECAs in spherical FTVR displays, we are interested in how pointing gestures in combination with verbal cues can help to establish a joint attention in a real world assembly scenario.

An early example of how pointing can be implemented in ECAs is Rae [11], a real estate agent using iconic, metaphoric, and deictic gestures. Rae uses pointing to indicate or emphasize objects in its virtual environment, such as features of homes, either complementary to speech or fully redundant [10]. Kopp et al. [22] designed Max, a human-size agent for cooperative construction tasks in a Collaborative Virtual Environment (CAVE). The agent employs speech, gaze, facial expressions, and gesture to guide the user through construction tasks.

While previous examples point in the virtual world only, MACK is an example of an ECA in mixed reality. The agent gives location directions and answers questions by using a combination of speech, gestures, and pointing into the real world, to the paper map in front of the user or to its surroundings to support voice directions [13]. Another example of an agent pointing in MR was presented by Anabuki et al. [1]. They created Welbo, a human-like robot agent that helps users in an MR living room. In the living room, users can interact with objects and simulate virtual furniture in the physical space. Welbo has the ability to have conversations with users and react to their instructions by moving furniture and guiding users with pointing gestures. These examples show the promise that ECAs have for pointing in MR spaces. However, additional research is needed to examine if pointing in MR space improves the interaction with ECAs. In our user study, we investigate the interaction with an ECA pointing from within virtuality to reality in an assembly scenario.

A novel approach to guide attention towards distant objects by using gaze was presented by Otsuki et al. [27]. To support remote collaborative tasks, they created "ThirdEye", a hemispherical display that shows tracked eye movement of remote participants. In a user study Otsuki et al. [27] showed that ThirdEye can lead the observer's attention to objects faster compared to only showing the image of the remote participant's face. The results underline the importance of using additional gaze cues for leading attention for remote collaborative tasks. Following this result, we include in addition to pointing cues.

All these examples show how ECAs, much like humans, are able

to use gestures and gaze to enable more natural interactions and help in completing tasks. With respect to user testing, previous work already showed good pointing accuracy of ECAs in a spherical FTVR displays. In reality, people do not rely on pointing gestures exclusively [5]. Thus, we evaluated the interaction experience of an ECA with pointing gestures in a real world assembly scenario in combination with voice.

## 2.4 Assembly Instructions

The most commonly used method for assembly instructions is the traditional paper manual. While a paper manual can show explanations in combination with pictures of the model status for each step, it does not help in identifying similar pieces and viewing the steps in 3D or from different viewpoints. Previous work has already presented different approaches for improving assembly instructions through technology. Two AR approaches were presented by Blattgerste et al. [7]. The first approach is to display the 2D images of the paper instructions into the user's field of view. The second approach uses in-situ instructions to overlay a marker for piece identification as well as a virtual model of the piece at the correct assembly position using AR glasses or smartphones. Their study results suggest a combination of in-situ feedback for picking the correct piece and pictorial feedback for assembly. Instead of using in-situ feedback for piece identification, our ECA points to the pieces. Based on the shown importance of pictorial feedback for assembly, we include a 3D model as assembly help.

To enable more helpful visual instructions for assembly, Yamaguchi et al. [36] presented a novel approach for generating and visualizing 3D AR tutorials with viewpoint control at runtime. The instructions are shown in an AR "magic mirror" display, which aligns the user's viewpoint of the physical object with the virtual 3D instructions. While the results of their user study did not show significant differences in task completion time and number of errors compared to traditional video tutorials, the AR mirror system led to significantly less mental effort. Subjective results also demonstrated the advantages of the system.

Another possibility to guide people through an assembly process is by using an agent. As described above, an example is Max, an ECA using pointing and other gestures to indicate virtual pieces and collaborate with users in the assembly of a virtual model in a CAVE [22]. In contrast to Max, our ECA points from within a FTVR display to real pieces with the goal of guiding users through a real world assembly process. We use an assembly task to investigate our ECA with pointing gestures, since assembly tasks require piece identification and thus, pointing is especially helpful. We focus on using pointing gestures for piece identification, but provide a virtual 3D model in front of the avatar as an assembly help since pictorial feedback for assembly was shown as most helpful [7].

## 3 DESIGN FACTORS

This section provides a description of the key aspects of our ECA design and implementation, including display form, appearance, speech, gestures and virtual model.

## 3.1 Display Form

We chose to use a spherical FTVR display for our ECA. Since FTVR displays are situated in MR space, compared to immersive Virtual Reality (VR) displays, they can enable pointing from within the display to real objects surrounding it. FTVR displays, introduced by Ware et al. [33], have been shown to increase the perception of three dimensionality of virtual objects. Motion parallax and stereoscopic cues are essential for interpreting pointing gestures and therefore FTVR displays, which provide these cues and create spatial 3D effects by rendering perspective-corrected vision, are particularly suitable for pointing [20]. Spherical FTVR displays improve depth and size perception compared to flat FTVR displays, hence are a

Table 1: Example voice instruction for both the showing pieces and pointing ECA for both steps, indicating a piece and explaining the assembly.

|  | Showing Pieces | Pointing |
|---|---|---|
| Indicate a Piece | "Take this blue screw" | "Take that blue screw" |
| Explain Assembly | "Use it to attach the 2 black connectors to the left yellow tube" | "Use it to attach the 2 black connectors to that yellow tube" |

more suitable form of FTVR for interpreting pointing targets [38]. Previous research already compared spherical FTVR displays to flat FTVR displays to illustrate improved performance. Thus, we compare the spherical FTVR display to a traditional flat 2D screen, as used in current state-of-the-art home assistants.

## 3.2 ECA Appearance

Human-like representations of ECAs are subject to the uncanny valley effect, which occurs when ECAs mimic human features in too much detail, while not fully succeeding, so that they appear unnatural, with an even bigger effect when movement is added [25]. Therefore we decided to use a female Japanese cartoon character with human-like traits while keeping a non-human appearance, as suggested by Schneider et al. [28]. The ECA we used has non-human proportions with big eyes a small nose and mouth. Considering the limited display size and the fact that our assembly task only requires seeing the upper body, Yoon et al. [37] suggest using a half body avatar. We scaled our upper body ECA as big as possible to improve gesture perception while allowing to extend the arm completely for pointing in both displays. This is in accordance with the use case of ECAs since, even though prototypes for life-size displays exist in research [20], in practice display sizes of home assistants are relatively small.

We implemented an idle animation state that is played in a infinite loop and consists of subtle arm and upper body movements, to make the ECA appear more active and alive [18]. To make our ECA feel more vivid, we added a blinking animation with random blinks of a rate between 3-5s, following the findings of Takashima et al. [29].

## 3.3 Speech

For the ECA's speech we used IBM's Watson Text-to-Speech (TTS) to generate verbal instructions for each assembly step from written text. We used the Oculus LipSync asset to match lip movement with spoken utterances. The asset uses blend shapes included in the avatar model to animate lip movement accordingly.

In every assembly step, the ECA first indicates which piece is needed, followed by a description of where the piece has to be attached. In the first step, the ECA broadly describes a piece accompanied by either a pointing gesture or showing a virtual piece by holding it up. In the second step, the ECA either gives broad verbal cues while pointing to the target position or only gives more detailed voice instructions explaining where to attach the piece. The pure voice version describes where the parts need to go in a more detailed manner than the voice in the pointing-added version to help the user complete the task with similar aid level. This is suggested by the substitution hypothesis of Bangerter et al. [4]. Voice instructions for an example assembly step for both the pointing and showing pieces ECA are shown in Table 1.

## 3.4 Gestures

To examine an assembly assistant pointing into the real world, two different gestures, pointing and showing pieces by holding them up,

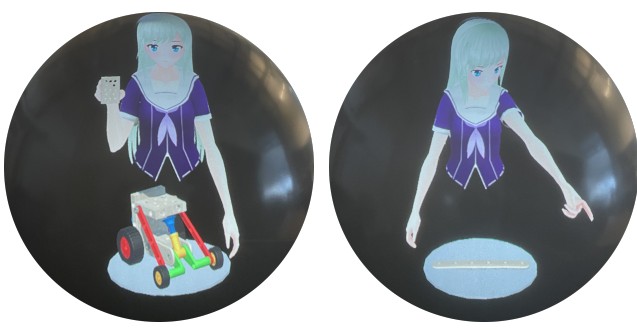

Figure 2: ECA showing a virtual piece (left) and pointing (right) in the spherical FTVR display. The virtual model state is displayed in front of the ECA.

were implemented. Both gestures accompany a voice instruction and substitute a spatial location expression of a piece.

Pointing

Previous research suggests that hand gestures combined with head rotation provide the highest accuracy and naturalness compared to hand or head only cues, especially for fine pointing [34]. Thus, we implemented hand as well as head animations to facilitate distinction between close pointing targets. Humans point by aligning their fingertip with the gaze of their dominant eye, while the observer interprets the pointing gesture by referring to the pointer's arm vector [5]. This might lead to ambiguity because the target interpreted by the observer is different from the actual intended pointing target by the pointer followed by the eye-fingertip line. Wu et al. [35] showed that using arm vector pointing for virtual avatars provides pointing with comparable and in some cases better accuracy compared to the pointing of a real person. Therefore our ECA uses arm vector pointing by outstretching the arm and index finger as well as rotating the head towards the target without eye-fingertip alignment (see Figure 2).

We implemented the pointing animation in Unity3D using inverse kinematics (IK) to enable the ECA to adapt the pointing animation to variable pointing targets during runtime. This allows a natural-looking arm raise animation while implementing a variable end position where the ECA's arm is outstretched, by building a vector from the shoulder to the index finger and towards the distant target. Instead of using object recognition, we decided to run a Wizard of Oz experiment to avoid recognition errors. Eye movement was not included, since testing revealed that there was no recognizable difference due to the big cartoon style eyes, which were always looking like they would face the target when the head was rotated towards it.

Showing Pieces

As a comparison to the pointing ECA, we also implemented a showing animation, where the ECA holds up virtual pieces instead of pointing to physical pieces in the real world (see Figure 2). The virtual pieces were created by measuring the physical Brio Builder pieces and modeling a virtual representation of them using Blender. The main animation was created using a video of a person holding a piece up as a reference and adding keystrokes to reconstruct the motion for the avatar.

### 3.5 Virtual Model

In front of the ECA, we displayed the model state after each assembly step on a small table that is floating in front of the avatar (see Figure 2). In a small pilot trial, we first tested the system without an additional visual representation of the model. The trial showed that it is very difficult to complete an assembly task without a visual aid

Table 2: Overview of the five conditions of the user study.

|  | Showing Pieces | Pointing |
| --- | --- | --- |
| Flat (2D) | X | X |
| Spherical FTVR (3D) | X | X |
| Paper Manual | - | - |

while relying on voice instructions and gestures only, especially because humans are used to rely on visual aids, like paper manuals, for assembly tasks. Thus, we decided to provide a virtual representation of the model state, allowing participants to verify if they picked the right piece, as well as give an additional visual aid for the assembly. In order to prevent participants from picking a piece based on the virtual model instead of the pointing or showing cue and therefore having a confounding influence on the study results, we displayed the model state only after the piece indication step, while the ECA explains the assembly (see Table 1).

### 4 EXPERIMENT

The goal of our experiment is to investigate the effect of our ECA with pointing gestures in an assembly scenario. We compare our pointing ECA in a spherical FTVR display to the same ECA in a traditional flat display. To provide a fairer comparison for the flat 2D display we decided to include a condition that is more optimized for the flat display: an ECA holding up virtual pieces in front of its body. With a paper manual as baseline, we measured assembly task completion time, errors and the interaction experience. The five conditions are shown in Table 2.

### 4.1 Participants

Fifteen paid participants (7 male and 8 female) aged between 18 and 45 were recruited from a local university to participate with a compensation of $10. All participants had normal or corrected to normal vision. None of them used Brio Builder construction sets before.

### 4.2 Apparatus

We used a 30cm diameter spherical FTVR and a flat display to conduct the experiment. To create a 360°image, four Optoma GT750ST stereo projectors with a 1024 x 768 pixel resolution and a frame rate of 120hz rear project onto the spherical surface, making a total NVIDIA Mosaic resolution of 4096x768 at 34.58 ppi [16]. A computer equipped with a NVIDIA Quadro K5200 graphics card runs the Unity application and sends the rendering content to all four projectors. We adopted an automated camera-based multi-projector calibration technique [39], to enable a seamless image with 1-2 millimeter accuracy. NVIDIA Mosaic synchronizes all screens in resolution and frame rate for stereo rendering and enables synchronization of XPand RF shutter glasses to generate stereo images with 60hz for each eye. The total latency lies between 10-20msec [16]. The OptiTrack optical tracking system was used for head tracking by attaching passive markers to the shutter glasses. To adapt the viewpoint to each participant, we used a pattern-based viewpoint calibration [31] with an average error of less than 1°. The spherical FTVR provides depth cues such as stereoscopic cues and motion parallax.

For the flat display condition, we also used an Optoma GT750ST projector with the same 1024x768 pixel resolution and 120Hz frame rate to rear-project on a flat screen to minimize differences between the flat and spherical display. The flat display's physical screen size is 36cm x 27cm which results in a similar screen area as the spherical screen with a 30cm diameter. In contrast to the spherical screen, the flat screen does not provide motion parallax, stereo rendering or perspective corrected images.

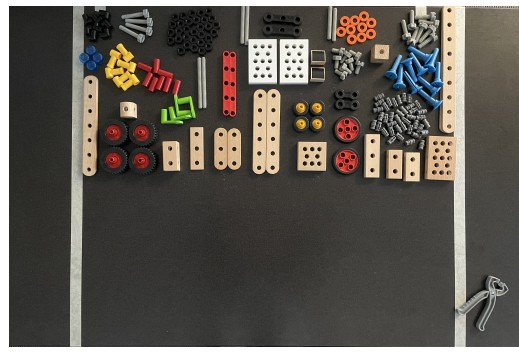

Figure 3: Top view of the table used in the experiment where the pieces were laid out. The free space was used for the assembly.

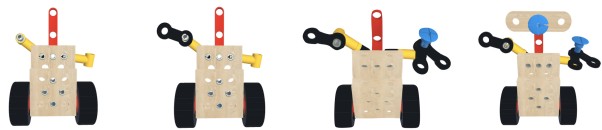

Figure 4: Extract of the paper manual used in the study, showing four assembly steps.

The physical Brio Builder pieces were laid out on a table within a marked area of the size 83x76 cm, as close to the ECA as possible, since the detection accuracy of pointing gestures decreases in distance [23]. All pieces were laid out in the same layout for all conditions and participants, to minimize differences. The same table was used for all the conditions. The study setup with all pieces laid out on the table is shown in Figure 3. In front of the pieces, there was free space where participants assembled the model. Both the spherical and flat display was placed so that the perceived size and distance of the avatar is similar.

We developed a Unity3D application for the experiment to animate and render the ECA and record task completion time. Our ECA was based on [2], and the virtual Brio Builder pieces used in the application were modeled using Blender. For the paper manual, we used the same models as shown virtually in the ECA conditions. The paper manual was color printed single-sided on large (11x17″) paper (see Figure 4).

### 4.3 Design

The experiment was conducted using a 2 x 2 within-subjects factorial design with a baseline paper manual condition:

- **C1** Display Form: spherical FTVR display (3D) or flat display (2D).

- **C2** Gesture: pointing (P) or holding a piece up (H).

For every condition, we used a different model, resulting in 5 models used throughout the experiment each consisting of 30 pieces (see Figure 5). The combination of display form/gesture and model as well as the sequence of conditions was fully counterbalanced using Latin squares. For quantitative analysis, we measured task completion time and errors. We collected subjective data about the interaction experience through a questionnaire. Furthermore, we measured the perceived workload using the raw Nasa TLX [17].

### 4.4 Procedure

First, we asked participants to sign a consent form and fill in a demographic questionnaire. We then explained the procedure of

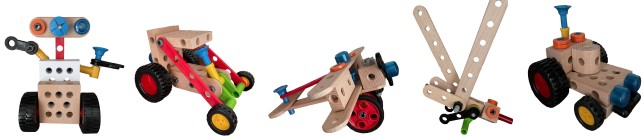

Figure 5: Photos of the five physical models that were assembled in the study.

the study and guided them through a viewpoint calibration. Each participant performed every condition once: two different display form factors combined with two different gesture types and a paper manual as a baseline, resulting in five assembly rounds per participant. Participants were asked to stand in front of the table with the laid-out pieces. In the paper manual condition, participants were instructed to follow the assembly steps shown on the images. They were allowed to navigate through the manual in their own pace and if needed jump back to previous pages, as they would naturally use a paper manual by themselves.

In the assembly assistant conditions, participants were instructed to follow the instructions given by the ECA. Participants were instructed to always pick a piece after the indication step and were allowed to change the piece in the next step if they later notice that they picked a wrong one. They were also allowed to move freely around the table during the assembly process. Each assembly step started with the ECA showing a piece or pointing at a piece required for the following assembly step accompanied by a verbal cue. Once participants decided for a piece, the avatar either only explains the next step or explains and points at the assembly position. At the same time, the model state is shown in front of the avatar as seen in Figure 2. Once the ECA received a verbal response, the next assembly step is started. It took about 5-10 minutes to complete one model assembly.

At the end of each assembly round, we presented twelve five-level Likert scale questions to participants and asked them to rate each in the range between "strongly disagree" and "strongly agree". The questions addressed character behavior, presence, and perception as well as general questions about the experience. After the paper manual round, participants were only asked to answer the four general experience questions.

Once participants completed the entire experiment, they filled out an overall questionnaire. They were asked to rate and explain which display form they prefer for both the showing pieces condition and the pointing condition. Additionally, they were asked to rank the instruction modes: paper manual, showing pieces and pointing and specify reasons for their preference. The entire experiment took about 60 minutes.

### 4.5 Results

In the following section, we describe the findings of our user study regarding work load, assembly completion time, errors and user experience.

#### 4.5.1 Work Load

First, we analyzed the raw TLX score over the different rounds to determine if potential work load or fatigue effects had to be considered in the further analysis. The mean raw TLX score was $M = 27.0$ $(SD = 14.5)$ after the first, $M = 33.9$ $(SD = 18.9)$ after the second, $M = 25.9$ $(SD = 14.0)$ after the third, $M = 26.3$ $(SD = 12.1)$ after the fourth and $M = 24.4$ $(SD = 13.5)$ after the last assembly round. A RM-ANOVA was conducted to reveal if the order significantly influenced the work load. The analysis did not reveal a significant effect of assembly round on work load $(F(4,56) = 1.848, p = .133)$. Therefore, we assume that effects on the assembly performance caused by work load or fatigue are negligible.

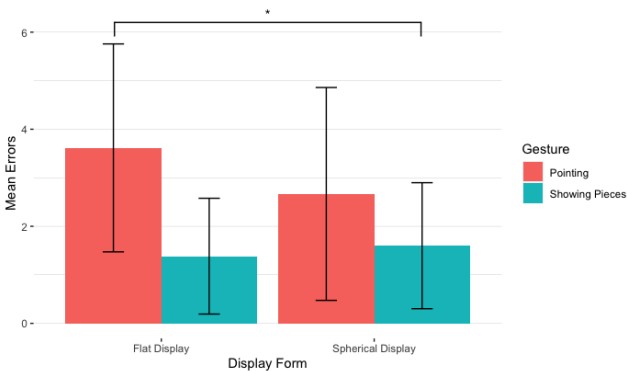

Figure 6: Recorded piece identification errors for all four ECA conditions with medians and 95% CIs. Significant values are reported in brackets for $p < .05$ (*).

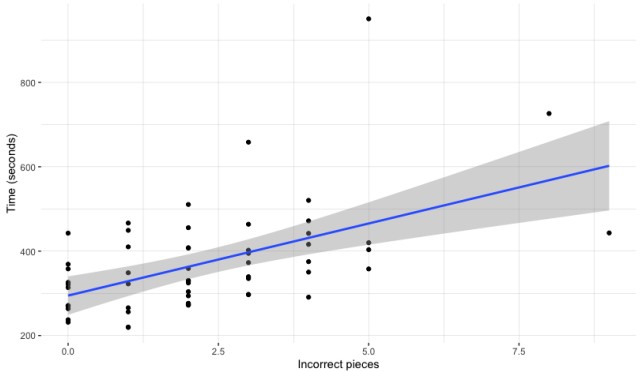

Figure 7: Correlation between number of incorrectly chosen pieces in each ECA assembly round and assembly task completion time in seconds.

We also analyzed all sub categories of the raw TLX using a RM-ANOVA. The only category for which a significant difference between conditions was found is frustration ($F(4,56) = 4.054, p < .01$). A two tailed t-test revealed the ECA pointing in the flat display ($M = 42.0, SD = 26.9$) led to a significantly higher frustration rating than the ECA that was holding up pieces in the spherical display ($t = -2.598, p < .05$), as well as the paper manual ($t = 2.327, p < .05$). There were no significant differences across remaining conditions.

### 4.5.2 Time

We measured task completion time for every condition. We performed a RM-ANOVA and found that instruction mode had no significant effect on assembly time ($F(4,56) = 0.816, p = 0.521$).

### 4.5.3 Piece Identification Errors

During the assembly process, errors were recorded and categorized in piece identification (finding the right piece) and assembly errors. The piece identification error includes wrongly picked pieces after the ECA was referring to them by showing a piece or pointing, including pieces that were corrected in the next assembly step. Since participants were able to see the model state right away in the paper manual condition and there was no separate piece identification step, the paper manual is not included in the piece identification statistics.

Results of the RM-ANOVA show a significant difference between conditions ($F(3,38) = 4.174, p < .05$). A two tailed t-test revealed that piece identification error was significantly lower ($t = -3.057, p < .01$) when the ECA was holding up pieces in the spherical display ($M = 1.6, SD = 1.3$) compared to when the ECA was pointing in the flat display ($M = 3.6, SD = 2.1$). There was no significant difference across remaining conditions.

### 4.5.4 Assembly Errors

Assembly errors were calculated by counting each incorrectly chosen and not corrected piece as well as wrongly attached pieces (e.g. pieces attached to a wrong hole or incorrectly rotated). The RM-ANOVA did not show a significant difference for the assembly errors between conditions ($F(4,52) = 0.640, p = .636$).

### 4.5.5 Time and Piece Identification Error Correlation

A Pearson correlation coefficient test was conducted and found a moderate positive correlation between assembly completion time and number of incorrectly identified pieces ($r(54) = 523, p < .001$). A visualization of the correlation can be found in Figure 7. As there was no separate identification step in the paper manual condition, target identification errors were only analyzed for the ECA conditions

|  |  | Flat (2D) | | Spherical (3D) | |
|---|---|---|---|---|---|
| **Statements** | **PM** | **H** | **P** | **H** | **P** |
| Felt like ECA was present | - | 2.4 (1.2) | 2.1 (0.9) | **3.0** (1.2) | 2.8 (0.8) |
| Correct piece identification | - | 3.5 (1.2) | 1.8 (0.9) | **3.9** (1.2) | 2.2 (0.9) |
| Enjoyed display form | - | 3.3 (0.9) | 2.9 (1.0) | **4.2** (0.8) | 3.8 (0.9) |
| ECA / manual was helpful | **4.2** (0.8) | 3.9 (0.9) | 3.0 (0.9) | 4.1 (0.8) | 4.0 (0.7) |
| Easy to follow steps | 3.7 (1.1) | 4.1 (0.6) | 2.1 (1.1) | **4.4** (0.9) | 2.7 (0.9) |
| Liked gesture / manual | 3.9 (1.1) | 4.1 (0.7) | 2.8 (1.1) | **4.3** (1.1) | 3.2 (0.8) |

Table 3: Mean and standard deviation of significant questionnaire responses for all five conditions: paper manual (PM), holding pieces up (H) and pointing (P) for the spherical and flat display. Higher scores indicate stronger agreement ranging from 1 (strongly disagree) to 5 (strongly agree).

and therefore paper manual times are not included in the correlation analysis.

### 4.5.6 Subjective Ratings

A Friedman ranked sum test was performed on all twelve five-level Likert scale questions. Participants rated each in the range between 1 ("strongly disagree") and 5 ("strongly agree"). The first eight questions which addressed character behavior, character presence and perception were only asked after the four ECA conditions. For all significant statements, mean and standard deviation values are shown in Table 3.

**Realism of Gestures** The Friedman ranked sum test did not reveal a significant difference between conditions for realism of gestures ($X^2(3) = 3.44, p = .329$), speech ($X^2(3) = 5.64, p = .130$) and fidelity ($X^2(3) = 3.660, p = .301$). There was also no difference between conditions for the statement that "gestures made ECA seem more realistic" ($X^2(3) = 3.74, p = .291$) and that "gestures strengthen the connection" between the ECA and themselves ($X^2(3) = 5.640, p = .130$).

**ECA Presence** The statement "I felt like ECA was present in the

real world" was rated significantly different between conditions, as shown by the Friedman ranked sum test ($X^2(3) = 8.060, p < .05$). Post hoc analysis with Wilcoxon signed-rank tests for multiple comparisons resulted in a significant higher rating of the presence for both spherical conditions H ($z = 2.223, p < .05$) and P ($z = 1.988, p < .05$) compared to 2D-P. No significant differences were found for remaining pairs.

**Target Identification Confidence Level** A Friedman ranked sum test revealed a significant difference across conditions regarding confidence level for target identification ($X^2(3) = 21.460, p < .001$). Post-hoc analysis with Wilcoxon signed rank test shows a significant effect between 3D-H and 3D-P ($z = 3.076, p < .01$) or 2D-P ($z = 3.180, p < .01$). It also revealed a significant effect between 2D-H and 3D-P ($z = 2.667, p < .01$) or 2D-P ($z = 3.040, p < .01$). No significant difference was found for remaining pairs.

**Assembly Confidence Level** The participants' confidence level of correct assembly did not show a significant difference between conditions ($X2(4) = 7.747, p = .101$).

**Enjoyment of Display Form** The enjoyment was rated significantly different as revealed by a Friedman ranked sum test ($X^2(3) = 11.340, p < .05$). Post-hoc analysis with Wilcoxon signed rank indicated higher enjoyment of 3D-H compared to 2D-H ($z = 2.934, p < .01$) and 2D-P ($z = 2.497, p < .05$). There was no significant effect between remaining conditions.

**Helpfulness** The Friedman ranked sum test revealed a significant difference for helpfulness ($X^2(4) = 11.800, p < .05$). Results of the post-hoc analysis with Wilcoxon signed rank test revealed that 2D-P was significantly less helpful than the other four instruction modes 3D-H ($z = 2.548, p < .05$), 3D-P ($z = 2.623, p < .01$), 2D-H ($z = 2.578, p = .01$) and PM ($z = 2.785, p = .01$). Results showed no significant differences between remaining conditions.

**Easy to Follow** For the statement "It was easy to follow the assembly steps", the Friedman ranked sum test revealed a significant difference ($X^2(4) = 30.160, p < .001$). Post-hoc Wilcoxon signed rank indicated that it is significantly more difficult for 3D-P than for 3D-H ($z = 3.076, p < .01$) or 2D-H ($z = 2.934, p < .01$) and PM ($z = 2.192, p < .05$). It was also significantly harder to follow the instructions for 2D-P compared 3D-H ($z = 3.408, p < .001$) or 2D-H ($z = 3.296, p < .001$) as well as PM ($z = 2.803, p < .01$). No significant difference was found across remaining conditions.

**General Preference** For participants' preference, the Friedman ranked sum test revealed a significant difference ($X^2(4) = 20.680, p < .001$). Post-hoc Wilcoxon signed rank showed that 3D-H was liked significantly more than 3D-P ($z = 2.934, p < .01$) and 2D-P ($z = 3.060, p < .01$). 2D-H was also liked significantly more than 2D-P ($z = 2.934, p < .01$). Moreover PM was liked significantly more than 2D-P ($z = 2.079, p < .05$). There were no significant differences between remaining conditions.

### 4.5.7 Overall Ratings

For holding pieces up conditions, most participants preferred the spherical FTVR display (73.3%) over the flat 2D display. 26.7% of participants preferred the 2D display. For pointing, the spherical FTVR display was preferred by 40.0%, while 6.67% preferred 2D. 53.3% of participants indicated that there was no difference between displays. The display rating results are shown in Figure 8.

In the overall rating of instructions mode, the ECA that was holding pieces up was ranked first by 53.3% of participants and

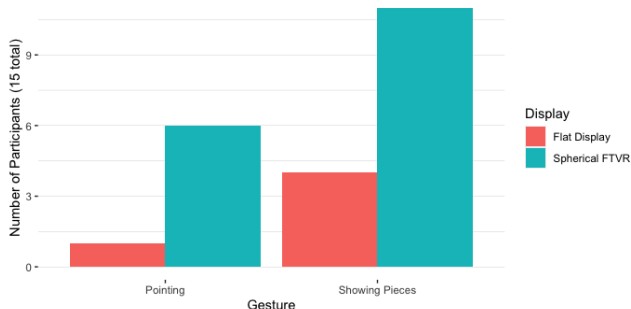

Figure 8: Overall rating of the display forms flat and spherical FTVR for the conditions pointing and showing pieces.

second by 46.67% of participants. The paper manual was ranked first as well as second by 40.0% and third by 20.0%. 6.67% of participants ranked the pointing ECA first, 13.3% second and 80.0% of participants third.

## 5 DISCUSSION

In this section, we discuss the three design factors display form, ECA appearance, speech and gesture to provide interpretations of our findings.

### 5.1 Reflection on Design Factors

#### 5.1.1 ECA Appearance

The subjective feedback regarding the ECA's appearance showed no significant differences between conditions, which is not surprising as the displayed ECA was the same in all four conditions. Nevertheless, results show how the ECA design was rated overall. Participants rated the realism of the ECA's movement and speech neutral. Even though we chose a not too human-like avatar to prevent the uncanny valley effect [25], it is particularly difficult to animate movements and implement speech without causing an unnatural appearance. Participants did not express comments regarding the character model and its gestures in the overall questionnaire.

#### 5.1.2 Gestures

In our study, we compared an ECA that is showing virtual pieces to a pointing ECA. Results show that instruction mode did significantly affect assembly time, which is composed of listening to a speech instruction, choosing a piece, and attaching the piece. The listening time was similar for each participant, while the duration of choosing a piece and attaching it differed. Thus, piece identification errors led to longer task completion times when participants had to replace a wrong piece for the correct one. This is in accordance with the correlation between assembly task completion time and the number of incorrectly chosen pieces (see Figure 7). It was observed that when the ECA was pointing at pieces that were unambiguous to identify, participants were much faster in picking the piece compared to other conditions, as they did not have to search on the whole table. But since pointing led to more piece identification errors, the faster piece identification for some pieces did not lead to a shorter assembly completion time for pointing compared to the other conditions.

Error data shows a significantly higher piece identification accuracy for showing virtual pieces in the spherical FTVR display than for pointing in the flat display. Although there was no significant difference found between remaining conditions, results hint that showing the virtual pieces resulted in a lower piece identification error rate than pointing towards them (see Figure 6). It was observed that many participants had difficulties finding the right pointing targets, when voice alone was ambiguous and ambiguous pieces were

placed side by side. This might be caused by multiple factors. First, the pointing targets were laid out closely together and in multiple lines in front of the ECA. This could have caused worse detection results than in previous studies, where the near far dimension was not investigated [34]. Second, we observed that participants lost trust in the ECA once they identified a wrong piece and had to correct it. While a high pointing accuracy of 82.6% was shown in a previous study which was conducted using a similar ECA with arm-vector pointing [34], this accuracy might not be high enough in an assembly scenario. Even though participants identified most of the 30 pieces correctly, they seemed discouraged after they chose a wrong one, which is also reflected in the confidence participants indicated in the questionnaire and the ranking results. 7 of 15 participants preferred showing virtual pieces, and 6 of 15 participants preferred the paper manual over pointing.

### 5.1.3 Display Form

In the flat display conditions, participants had difficulties interpreting the correct pointing targets of the ECA which led to a higher target identification error. This might be caused by the lack of depth cues, which helped to detect the correct pointing target in the spherical FTVR display. It was observed that some participants were moving around the display while the ECA was pointing, which seemed to make the piece identification easier (see Figure 9). That is in line with previous research which showed that left and right areas, where participants see the arm pointing away from them, are more prone to misjudgments, while the front area was less difficult to recognize [34]. When the ECA was for example pointing to the right side and participants were moving towards the pointing target, they might be able to identify the target more accurately because the ECA is then pointing towards them. The same difference in behavior also applies to viewing the virtual model, which was visible in 360°view, when participants moved around.

We were surprised to find that while 11 participants preferred the spherical FTVR display when the ECA was showing pieces, only 6 participants preferred the spherical FTVR display for pointing. Eight participants reported no difference, although target identification errors were lower with the spherical display. Participants who preferred the spherical FTVR display noted that they felt like it was more accurate and easier to interpret the pointing targets, while participants that answered with "no difference" had difficulties detecting pointing targets in general so they had no preference. That is surprising, as the spherical FTVR display provides more depth cues and led to fewer errors. A possible explanation is that participants who were not confident in identifying pointing targets got discouraged resulting in feedback like "pointing is inaccurate in general" and the rating "no difference" between display forms, even though the spherical FTVR resulted in fewer errors.

There was a large difference between participants and their behavior interacting with the spherical FTVR display in general. Some participants moved around the display more and therefore have taken more advantage of depth cues and the possibility to get different perspectives of the ECA, the pieces and the displayed model. Others did not move at all, although all participants received the same instructions in the beginning. Therefore there was a smaller difference between both displays for participants who were standing at the same position during the whole assembly round.

### 5.2 Comparison to Paper Manual

The recorded assembly errors were below one in all conditions and therefore differences were not significant, even though piece identification errors were much larger. This shows that participants recognized incorrect pieces when the model state including the previously chosen piece was shown and thus corrected the piece, leading to a correctly assembled model. It was surprising to find that there was no significant difference in the assembly error between

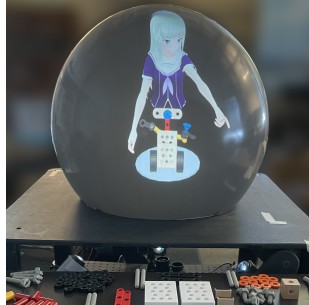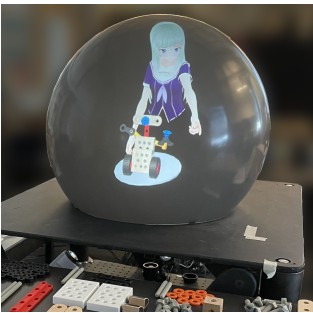

Figure 9: The perspective change when moving around the display might help to identify pointing targets more accurately. Here, the observer was moving from the front position (left) to the right side (right).

conditions, as the paper manual and the flat display did not provide a side or back view of the virtual model, making it "hard to see the other side of the model", as noted by participants. Others mentioned that the "virtual 3D model is always more helpful than paper manual" and "paper manual needs detail attention". Nevertheless, participants were able to assemble the model as correctly with the paper manual as when using the ECA. A possible reason could be that the paper manual allowed participants to "[easily] go back multiple steps", which then also provided different perspectives of the models, as well as gave participants the possibility to see if they made a mistake before. Most participants were observed navigating back and forth through the manual during the assembly process. Another reason mentioned by participants is that they are more "used to a paper manual" and to "reading visual assembly scenarios". This similarity of paper manual and ECA is also reflected in the statement rankings for helpfulness and preference (see Table 3).

### 5.3 Design Implications

Our study revealed challenges when designing ECAs that point into the real world. While previous research found a high detection accuracy for ECAs pointing into the real world [34], this accuracy might not be high enough in an assembly scenario. We observed, that it is particularly important to reach a high target detection accuracy to avoid frustration in the assembly process. By using pointing, the ECA was able to guide the attention of participants to a broader region, which helped participants narrow down the number of possible pieces. However, when multiple similar pointing targets are located closely together, they were not able to identify the correct piece using the pointing cue only. Thus, we suggest to implement indirect methods for precision tasks, like for example displaying a virtual piece. An example of how 3D instructions can improve the assembly in combination with viewpoint control was presented by Yamaguchi et al. [36]. This could also be implemented in our spherical FTVR display, which allows for 3D view and viewpoint control. Since participants liked the ECA using pointing gestures in general, pointing could be implemented in addition to indirect methods.

### 6 LIMITATIONS AND FUTURE WORK

In the following, we discuss six limitations along with opportunities they present for future research. First, in our study we only used one construction set and pieces were arranged in a fixed layout on the table to increase comparability between participants and conditions. Thus, it would be interesting to conduct a similar study using different pieces, such as a real furniture construction set, arrange pieces in a different layout, or use pieces without prior arrangement.

Second, we observed that participants behaved very differently when using the spherical FTVR display. While some participants

used the additional cues, e.g. by moving around to get an additional perspective of the ECA or the virtual model, others did not move at all. Therefore it would be interesting to investigate whether more experience with the pointing ECA would improve the ability to identify correct pointing targets. A possibility could be to include a training phase where participants are animated to move around the display and identify pointing targets with feedback.

Third, while participants were able to detect pointing targets accurately when they were placed far enough apart or described detailed enough, it was difficult to distinguish closely located parts with broad voice descriptions. Since previous research showed that verbal descriptions should be substituted by gesture where possible instead of implementing both redundantly [4], it first requires future studies to quantify the detection accuracy for arm vector pointing to targets on a horizontal plane and determine at which distance targets get ambiguous.

Fourth, in our study we only compared the ECA's pointing to showing virtual pieces and a paper manual baseline. Results show a low piece identification error for the showing pieces ECA, even though the ECA was only holding the virtual pieces up. Therefore the question arises, whether an ECA does generally provide an advantage over a virtual model, especially because some participants noted that they felt pressured when using an ECA for the assembly in comparison to the paper manual. In contrast to a 2D paper manual, a 3D visualization displayed in a spherical FTVR display could provide depth cues. Future studies could investigate, whether an embodied human-like assistant provides an advantage over a 3D visualization of the assembly steps.

Fifth, our ECA only explained the assembly steps using voice and gestures. However, in an assembly scenario with a human assistant, people would not only follow the explanations, but also ask questions when they are unsure in an assembly step. Thus, a future step would be to implement a feedback mechanism and conduct further research to investigate, whether giving feedback would improve the error rate, assembly time and interaction experience.

Last, we only implemented deictic pointing gestures. Additionally, it would be possible to provide multiple gesture types, as they are used in human communication. An example was shown in previous research [22]. Their presented ECA used deictic pointing in combination with metaphoric gestures, to demonstrate how pieces should be placed, for example by crossing fingers to indicate that pieces have to be attached together in a 90 degree angle. Thus, future studies could investigate if the implementation of additional gestures enhances the interaction with ECAs.

## 7 CONCLUSIONS

In this paper, we presented an ECA with the ability to point into the real world to investigate, whether spherical FTVR displays affect the interpretation of the ECA's pointing gestures, as well as examine the effect of ECAs with pointing gestures in an assembly scenario in general. We conducted a study to compare the pointing ECA in the spherical FTVR display to an ECA holding up virtual pieces as well as to the same ECAs in a flat display with a paper manual as baseline. Participants had to assemble different construction toy models while measuring assembly time, errors and user experience using a questionnaire.

Our results show that the spherical FTVR display had no significant effect on assembly time or errors, while it was preferred in all ECA conditions by participants and was shown to lead to a higher presence rating. The ECA with pointing gestures could not reduce assembly time or errors compared to the ECA that showed virtual pieces or the paper manual, though it was rated as helpful in the assembly process. Our findings show that pointing is helpful to guide attention to a broader region, but is not suitable for precise locations. For precise piece identification indirect methods, like showing the pieces, are more helpful and could be used in combina-

tion with direct methods, like pointing. These findings can be used to guide the design and development of ECAs that point into the real world, especially for assembly scenarios. Since home assistants are advancing in interaction possibilities, an ECA that provides gestures is expected to provide more natural human-like interactions and thus merge the boundaries between the virtual and real world.

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
