# OpenReview forum: "Attach That There: Investigating 3D Virtual Assembly Assistants That Point Into the Real World"
_graphicsinterface.org/Graphics_Interface/2023/Conference_SD — Submitted to GI 2023 - second deadline_

### Official Review · Reviewer_pmLX · 2023-04-21
**Investigating 3D Virtual Assembly Assistants using the Verbal and Pointing Gestures. Exploration of a novel, niche design. Clarity of presentation could be improved.**

**Rating:** 6
**Confidence:** 4

**Review:**

Pros:
The work contains a detailed analysis of the impact of receiving assistance from an ECA during an assembly task across 4 conditions plus a paper manual baseline.
This presents an interesting system in terms of potentially immersive ECA, which could be applied to different modalities like headsets.
Very thorough study with 5 tests, survey after each (with NASA TLX), timing and assembly errors recorded, with overall questionnaire and rankings at end.
The effect of an ECA using pointing into the real world accompanied by verbal cues on the interaction experience. (According to authors, no work has been done on this so far)
Comparison among five different conditions (showing/pointing pieces with different displays (flat 2D and 3D and a paper manual)  on 5 different physical assembly models using 15 participants seems reasonable.

Cons:
The narrative throughout the paper is predominantly focused on pointing, despite the results of the experiment showing that the display type was more important. This practice feels like the work is trying to twist the results to justify the outcome that was expected in the initial hypothesis.
The research area is somewhat niche and separated from the current standard, combining several design factors into one experiment. As the future work discusses, another baseline that could be explored is simply displaying a virtual model with voiceover instead of using an ECA.
No examples given of the 2D display condition.
Table 1 needs a stronger substitution hypothesis on why voice instructions need to be more detailed when showing pieces than when pointing to the pieces. Do the authors believe that showing the subject is less informative than pointing to it, and therefore, they added a more detailed voice description to the showing parts?
In section 4.5.2, it's better to remind the reader again what instruction modes (pointing, showing, paper manual?)  are and if this differs from display mode.  It's somehow confusing if the instruction mode included just verbal instruction or not.
Section 4.5.2 states that the type of instruction mode (such as pointing, showing, or using a paper manual) did not significantly impact assembly time. However, section 4.5.3 reports that there was a significantly lower rate of piece identification errors when using an ECA to hold up pieces in a spherical display. Does this mean that a spherical display or holding up a piece or both of them together have caused pieces to be identified easier? until here, one may conclude that just the display alone affected easier identification of the pieces.
Would it be preferable to report the effects of combining various instruction modes and displays to identify the optimal combination that yields the best overall assembly time and measure each condition's impact on assembly time?

A reader would expected to see a significant difference between the assembly error in the condition above (using an ECA to hold up pieces in a spherical display) and other conditions. The reason is that piece identification error ( finding the right piece) was significantly lower in this condition, and looking at the assembly error formula (counting each incorrectly chosen and not corrected piece as well as wrongly attached pieces), it could be concluded that since the piece identification error was low for the particular condition, then to have this condition’s assembly error with no difference to others (reported in section 4.5.4), we need to have more wrongly attached pieces (correctly identified but wrongly attached).
If this is the case, then the study should mention that “ECA holding up pieces in the spherical display correctly identified more pieces, but the rate of wrongly attaching was high, rendering this condition with no significant difference for the overall Assembly error to other conditions)
A reader might also expected to see less overall assembly error in using an ECA to hold up pieces in a spherical display, as according to Table 1, assembly explanation seems enough to help attach pieces in the right place.
In the Conclusion section, the author stated that the 3D display did not have a significant impact on assembly time and error. Additionally, they concluded that using an ECA to show pieces reduced assembly time. However, this contrasts with their earlier finding that the type of instruction mode (pointing, showing, or using a paper manual) did not significantly affect assembly time.(Section 4.5.2)



In the limitation section, Fourth limitation and the conclusion about it does not make sense or is not well explained.
For example it is mentioned that: “Results show a low piece identification error for the showing pieces ECA, even though the ECA was only holding the virtual pieces up. Therefore the question arises, whether an ECA does generally provide an advantage over a virtual model, especially because some participants noted that they felt pressured when using an ECA for the assembly in comparison to the paper manual. In contrast to a 2D paper manual, a 3D visualization displayed in a spherical FTVR display could provide depth cues. Future studies could investigate, whether an embodied human-like assistant provides an advantage over a 3D visualization of the assembly steps.”

On what basis is assembly performance defined, and why was it stated that assembly performance was enhanced when utilizing a spherical display to hold up pieces? Is there a connection between assembly performance to assembly error/time/piece identification error?
It was mentioned by the authors that there is no significant difference between assembly error in different conditions.

Quality:
Design choices for the ECA are well cited.
Implementation details are thorough for the design of the ECA, the tools used, and the objects that were assembled.

Clarity:
The narrative of the paper focuses predominantly on pointing, but the display type of flat vs. spherical was also an experimental condition. The related work motivates the choice of including a spherical display, but the paper should emphasize more that ECA in sphere vs. ECA in flat display was a research question.
The presentation of the results could be improved. Some results that seem important to the paper’s narrative are nested in paragraphs while results presented as insignificant receive figures, e.g., completion time.
The sentence beginning “While there are many studies on how humans perceive and use gestures…” on the first page seems to contradict itself. If humans perceive and use pointing gestures differently, then why wouldn’t this generalize to ECAs? It is a valid research question to confirm if it generalizes, but the claim in the sentence appears unsubstantiated.



General Comments:
“...distinguished in proximal and distal” should be “distinguished as proximal or distal”
On Page 4 ‘rely’ should be ‘relying’ in the following clause: “...especially because humans are used to rely on visual aids…”
It looks like there are only 4 conditions in Table 2 when the caption says 5.
NASA should be capitalized
- Mentioned Wizard of Oz demo (pg. 4) but then said pieces as close to screen as possible for detection accuracy (pg. 5); does that mean system detection (so using some computer vision) or user detecting what is pointed at?
- From the study design, it seems you will end up with a 5-column latin square (including the baseline paper condition and the display form w/ gesture method combinations).  With counterbalancing, this hopefully reduces sequential effects, but the paper mentions using 5 different models as well.  This gives me a couple sub-questions:
  - Same model for each one (e.g., paper baseline was always same model?) or different?
  - Are all 5 models of equivalent difficulty?  The difficulty measure need not be rigorous but it should be at least mentioned that they are comparable in some metric.
- The Pearson correlation r-statistic shared in 4.5.5 (pg. 6) seems to be a typo.  I would expect a correlation within -1 to +1.
- Smart quotes are turned backward throughout paper.  Likely a Latex formatting issue.
- Would be good to include an image of the flat display presentation.  It would help the reader get an understanding of what the user experience looked like for this condition.
- I like the future work idea of studying the movement interaction with the ECA more.  This could lead into what my preference for such a system would be: something running on an AR or VR headset with more immersive and interpretable behaviors.




Originality:
The work serves as an application of “pointing into the real world” which is a relatively new area.
Exploring the effectiveness of pointing gestures in assembly tasks is novel.


Significance:
The work does not strongly motivate why it is valuable to explore using gestures in assembly instructions other than the fact that humans use gestures in communication.
As the discussion points out, it is unclear if an ECA with pointing gestures has significant benefits over simply using a virtual model display.

---

### Official Review · Reviewer_Z52y · 2023-04-21
**Interesting study, but methodological errors.**

**Rating:** 4
**Confidence:** 4

**Review:**

This paper presents a study on how the presentation of a conversational agent impacts an assembly task. A study examines how fishtank vr compares with 2D projection, as well as how having the character hold, or point to an object impacts assembly. The study also includes a baseline condition of a paper manual. By analyzing completion time, errors, perceived task load, and subjective reports, the study finds the display has no effect on errors or completion time, but FTVR was preferred to the 2D projection. Pointing was found to be useful, but not in identifying precise location.

Overall, this paper was well written and easy to follow. The arguments were clear, and the paper situated itself well within prior work. The topic itself is interesting - ECA's are likely to become more prevalent as displays become more ubiquitous and AI continues to advance. Understanding whether different display modalities (e.g., FTVR) offer advantages over traditional displays is a useful endeavour.

My largest concerns with this paper are with the methodology and reporting of the results.

Regarding methodology, the number of measures examined seemed excessive and more in line with an exploratory study rather than a confirmatory study. Given the amount of tests run: ranking, rating,  TLX analysis (and sub-analysis), subjective questions it is not surprising that some tests were found significant. A more focused study with clearer hypotheses and narrower metrics would be better to truly understand the impacts of the different conditions quantitatively.

In addition, the wrong tests were reported in some cases. The RM-ANOVA described in 4.5.1 seems to be a 1x5 design, not a 2x2 design as described in 4.3. It should be very clearly stated what types of tests were used where. Given that TLX data is ordinal, non-parametric tests should be used. The post-hoc tests are incorrect. They should be Bonferroni-corrected paired t-tests, not just two-tailed t-tests. This likely impacts the significance of the results. Given that this paper is an empirical study, it is imperative that the stats be correctly documented.

Similarly, the reporting of all measures is crucial. If the measures are important enough to collect, they should be reported in detail. A graph for the time should be presented, even if it was not found to be significant. Without it, the reader doesn't have an idea of how long each condition took, the amount of noise, etc. Similar for all the metrics (e.g., assembly errors). This is especially true when there are not significant differences found through NHST. Was this because there was a lot of variability in the data, or because the sample size was not large enough? Presenting this graphically would be useful.

Concerns that didn't impact my rating:
$10 for an hour of work is quite low. It's unclear if this study was conducted in Canada or the US, but in the US, most states minimum wage is above $10, and in most Canadian provinces it is above $15. Living wages are higher than that. The methods should also mention whether the study was approved by an institutional review board.

Table 2 is confusing. Why is it presented in a table? It is unclear what the X and - represent within the table cell.

Overall, I think the study is interesting. I think the results of the study could be of value, but given the errors present and how they might impact the findings of the paper I cannot recommend it for publication at this time.

---

### Official Review · Reviewer_B4yi · 2023-04-25
**Interesting experiment, but needs more work ...**

**Rating:** 4
**Confidence:** 4

**Review:**

This paper is very interesting and while I believe that it is important to publish approaches that do not necessarily work well (not all papers have to show improvements), I would say the work is a little premature – this is mainly because of the clarity of writing/experiments themselves.
Possibly one of the main problems is that the paper is a little uneven, sometimes it’s explaining details that are not really necessary, sometimes they are skipping details that mean I am going to have to search up the information. Deictic gestures, for example, were badly explained (explained more as an example of use, rather than what they are) – and providing a reference doesn’t help, because it interrupts the flow of reading – in addition, the usage is muddled almost immediately (Deictic gestures are pointing gestures, so what is “Deictic pointing”? Pointing Gesture Pointing? It’s a small thing, but really interrupted the paper’s flow.
Another issue with the unevenness of the paper is that Table 2 seems really unnecessary (I mean it’s acceptable, but it doesn’t really clarify anything that wasn’t really understood); however, the results section was really dying for a table (the paragraph that starts “Easy to Follow”, on Page 7, was crying out for it to be a table (so were other paragraphs in this area), but it wasn’t … again, no table where there should have been tables, and tables where there shouldn’t be tables …
And when I got to 2D-P, I had to search back – too much abbreviation; which I eventually tracked to Table 3; it didn’t help that you used Spherical H and P and then went into 3D-H/3D-P later. Again a small thing, but really interrupted the flow of the paper.
Other areas, such as your Overall Ratings were very difficult to visual and Figure 8 didn’t really help (also the text was way too small for paper reading), this should have been reworked to make it more understandable. The problem is not the numbers (or the visualisation of the numbers), it’s the use of the numbers – for example a pie-chart would have been much better in this case (as the numbers should sum to 100% in most cases I believe).
Figure 3 was also not sufficient and would have been better to show all elements (the FTVR position, etc.) so it’s clear where everything is; it was not clear from your description.
Section 4.3 was really confusing, mainly because it was too brief and probably required a diagram – that was not included. I think I would have to sit down and put the experiment together to really understand the framework you put together – this made it difficult to assess as to its validity (again uneven reporting).
You also reported gender and age range, but never discussed the effect it had on the results, if any – based on your experimental setup in Section 4.3, I would say though that 15 participants didn’t really cover the necessary amount of information you should have collected to make a conclusion, because you essentially had 5 conditions, 5 models (of varying complexity) … the questionnaire may have been valid, but the analysis of times probably was too biased (again, I am not sure I understood it sufficiently from reading).
Apart from the reporting style, which needs to be significantly improved for publication, I found that your own review of prior art seems to suggest your method SHOULD have shown improvements – this seems to be the hypothesis you were working on; so at the time of publication (and with the many questions still in play) it seems that you had a problem that failed to prove that hypothesis and you just published the results. However, there are many questions that you should have asked, which means someone would have to repeat this experiment to verify the results. Things like “was the novelty distracting?”, “was the character too bland, lacking contrast?”, “where there any biases towards male or female participants?”, “was the age of the participant a factor?”, “Do we consider the BRIO manuals to be perfect?” (you did not show the manual, and I don’t know BRIO), “should you have compared to Lego?” (Brio is older than Lego, does this give more experience?), also was the direction of pointing an issue, “what eyesight test did you give at the beginning?”, “did you do a colour test?” …
Experiments like this need to be vigorous because they are easily invalidated by missed steps. Overall though it was interesting and with some key steps, would possibly be a very valid publication.